# Monitoring Cutaneous Leishmaniasis Lesions in Mice Undergoing Topical Miltefosine Treatment

**Laura Fernanda Neira [1],\*, Julio Cesar Mantilla [2] and Patricia Escobar [1],\***

[1] Centro de Investigación en Enfermedades Tropicales (CINTROP-UIS), Departamento de Ciencias Básicas, Escuela de Medicina, Universidad Industrial de Santander, Bucaramanga 680002, Colombia
[2] PAT-UIS, Departamento de Patología, Escuela de Medicina, Universidad Industrial de Santander, Bucaramanga 68002, Colombia; jumantil@uis.edu.co
\* Correspondence: laurafernandaneira@hotmail.com (L.F.N.); pescobar@uis.edu.co (P.E.)

**Abstract:** A study was conducted on BALB/c mice infected with *Leishmania (Leishmania) amazonensis* to analyse the effects of 0.5% miltefosine (MTF) hydrogel treatment on cutaneous leishmaniasis (CL) lesions. The mice were treated for 25 days topically, and lesion sizes, parasite loads, histopathology, ultrastructure, cytokines including interleukin 4 (IL-4), tumour necrosis factor alfa (TNFα), interferon gamma (IFNγ), IL-10, and vascular endothelial growth factor (VEGF) profiles were evaluated on days 0, 12, 25, and 85. After 12 days of treatment, the lesion sizes and parasite numbers decreased. By day 60 post treatment, there were no lesions and only a few parasites. At day 25, there was a temporary papillomatosis reaction, an increase in mast cells, a few giant cells, and granulomas, and a decrease in diffuse inflammatory infiltrate and parasites. Transmission electron microscopy (TEM) examination showed early ultrastructural changes, including macrophages without parasites and vacuoles containing electrodense material. At the different evaluated times, the cytokine regulation indexes (ICRs) decreased for IL-4, TNFα, and VEGF. According to the study, the 0.5% MTF hydrogel was effective and showed positive results from the early stages of usage. The MTF directly targeted parasites, downregulated the release of IL-4, TNFα, and VEGF, increased mast cell production, and induced granuloma reaction during evaluation periods.

**Keywords:** cutaneous leishmaniasis; cytokines; electron microscopy; histopathological patterns; topical miltefosine





## 1. Introduction

New World cutaneous leishmaniasis (NW-CL) is the most common form of leishmaniasis, averaging 52,645 cases annually in Latin America [1]. It is produced by an obligated intracellular protozoan parasite belonging to subgenus *Viannia* or *Leishmania*, transmitted to humans and reservoirs by infected female sandflies of the genus *Lutzomyia*. Localised NW-CL is the most common clinical form, characterised by skin lesions, including papules, nodules, or ulcerative volcano-like lesions that leave disfiguring scars. Differences in the number, shape, size, evolution time, and location of CL lesions, as well as their progression to mucosal or inactive disease, represent the heterogeneous clinical spectrum of NW-CL influenced by the host, *Leishmania* spp., vectors, and social and environmental factors [2–4]. After infection, various cells such as dendritic cells, natural killer cells, mast cells, neutrophils, and macrophages participate in parasite control through antigen recognition, migration, phagocytosis, and release of cytokines, chemokines, neutrophil extracellular traps, and reactive oxygen and nitrogen species (ROS and RNS) [3]. Cutaneous leishmaniasis control or progression is closely related to the types of macrophages and lymphocytes that result from immune response activation. Macrophages can differentiate into M1 cells, which produce proinflammatory cytokines, chemokines, RNS, and ROS (for disease control/Th1 response), or M2 macrophages, which produce interleukin 10 (IL-10)

and transforming growth factor beta (TGFβ) (for disease progression/Th2 response). In lymph nodes, infected dendritic cells activate naive T cells to differentiate into either Th1 cells secreting interferon gamma (IFNγ), tumour necrosis factor alfa (TNFα), and IL-12) or Th2 cells (releasing IL-4, IL-10, and IL-13) [5]. Achieving a balance between pro- and anti-inflammatory responses can result in moderate or even self-healing of CL [6]. However, an extreme polarisation towards Th1 or Th2 response can result in severe diseases, including mucocutaneous leishmaniasis, post-kala-azar dermal leishmaniasis (PKDL, in Old World visceral leishmaniasis), disseminated CL (DCL), and anergic diffuse CL [7]. From a histological view, different patterns with varying parasite loads can be seen under a microscope: one with macrophages, lymphocytes, and plasma cells and indistinct granulomas, and the other with well-defined granulomas. In addition, the epidermis may undergo pathological changes, including acanthosis, vasculitis, neuritis, necrosis, and fibrosis at varying intensities depending on healing, severity, or scarring [8,9].

Miltefosine (MTF) is a synthetic drug active against *Leishmania*, other protozoans, and neoplastic cells. It is used orally for all clinical forms of leishmaniasis [10]. In *Leishmania*, MTF inhibits phospholipid biosynthesis and mitochondrial cytochrome c oxidase and induces apoptosis-like cell death, disturbance of lipid-dependent cell signalling and alterations in the intracellular $Ca^{2+}$ homeostasis [11–14]. It acts as an immunomodulator, restoring responsiveness to IFNγ, boosting early IL-12 and TNFα production, regulating Th1/Th2 responses and increasing ROS production [15–18]. MTF treatment enhances macrophage function by inducing cytokine expression, upregulating cytokine receptors, and increasing phagocytic capacity and production of ROS and NO in cells and mice. It also increases $CD4^+$ and $CD8^+$ T cell proliferation and human oxidative burst [19–21]. MTF binds to albumin in plasma and accumulates in cell membranes' lipid rafts, increasing membrane fluidity, T-cell function and anti-IgE-induced histamine release from mast cells, and regulates eosinophilic inflammation [22,23].

We previously demonstrated the efficacy of 0.5% MTF hydrogel in treating CL in BALB/c mice infected with various NW *Leishmania* spp. [24]. MTF could positively affect the outcome of the disease, both as a leishmanicidal and as an immunomodulator. This paper aims to monitor the in-situ effects of 0.5% MTF hydrogel used topically on *Leishmania amazonensis*-infected mice during and after treatment. Some aspects of the relationship between parasites, host, and MTF were studied, including lesion evolution, parasite loads, ultrastructural changes, histopathological patterns, and cytokine modulation.

## 2. Materials and Methods

### 2.1. Reagents and Gel Preparation

Miltefosine was obtained from Cayman (Ann Arbor, MI, USA). Carbopol®940, sodium benzoate, and triethylamine were purchased from Sigma–Aldrich (St Louis, MO, USA). DMSO was purchased from Carlo Erba Reagenti (Rodano, Italy), and RPMI medium, Schneider's insect medium, and FCS were provided by Gibco (Grand Island, NY, USA). The preparation of 0.5% miltefosine hydrogel and vehicle (hydrogel without MTF) was described by Neira et al. [24].

### 2.2. Parasites

Promastigotes of *L. amazonensis* (MHOM/BR/73/LV78) were cultured in Schneider's medium plus 10% of heat-inactivated FCS (hiFCS) at 28 °C. Previously, the identification of parasite species was achieved through PCR-RFLP using *Leishmania (Viannia) braziliensis* (MHOM/BR/75/M2903), *Leishmania (Viannia) guyanensis* (MHOM/BR/75/M4147), *Leishmania infantum* (MHOM/BR/74/PP75) and *L. amazonensis* (IFLA/BR/67/PH8) reference strains kindly provided by Professor Silvane Fonseca Murta from the *Leishmania* Collection at the Rene Rochou Institute (FIOCRUZ-MINAS). The PCR amplifications targeted the heat shock protein 70 (HSP70) and internal transcribed spacer 1 (ITS1) [25,26]—Supplementary Figure S1.

### 2.3. Mice and Ethical Considerations

Female BALB/c mice aged 7–8 weeks were provided by the National Health Institute (NHI) in Bogotá, Colombia. The mice were housed under a 12 h light/dark cycle, with a temperature of 22 °C $\pm$ 2 °C and 55 $\pm$ 5% relative humidity. They had access to water and mouse food pellets ad libitum. The studies were performed according to the NHI Guide for the Care and Use of Laboratory Animals protocols approved by the Industrial University of Santander (Bucaramanga, Colombia) Ethics Committee (Act 23 of December 2019). Animal welfare was prioritised during the study. Mice were monitored for behavioural changes and signs of stress and weighed each week, before being humanely euthanised using a ketamine/xylazine mixture followed by cervical dislocation.

### 2.4. Infection and Treatment

The mice ($n$ = 40) were infected by subcutaneous injection of $5 \times 10^5$ stationary phase *L. amazonensis* promastigotes suspended in 100 µL of PBS, pH 7.2, in the shaven rump above the tail. When the lesions were >20 mm$^2$, they were randomly distributed into two experimental groups, one treated topically with 0.5% MTF hydrogel and the other with a vehicle directly over the CL lesion. The group treated with MTF had a sample size of $n$ = 24 mice, distributed in 4 subgroups of $n$ = 6 mice each. On the other hand, the vehicle group had a sample size of $n$ = 16 mice, distributed in 4 subgroups of $n$ = 4. Each subgroup represents the mice sacrificed at the different evaluated times The appropriate sample size (n) for our study was determined using G Power software (Statistical Power Analyses 3.1.9.4 for Windows), based on previously obtained values from our lab for standardizing tissue parasite loads and cytokine, with a power of 80% and alpha of 0.05.

Mice were sacrificed at four time points: day 0 (pre-treatment: $D_0$), day 12 (during treatment: $D_{12}$), day 25 (end of treatment: $D_{25}$), and day 60 post treatment (p.t.). Lesions were removed and evaluated for parasite loads, histopathologic and ultrastructure analysis, and cytokine levels.

### 2.5. Lesion Size Determination

Lesion sizes (LS) were measured at different time points (0, 12, 25, and 60 days p.t.) and compared to vehicle-treated mice. Wound dimensions were measured using a digital calliper, and the surface area was calculated by multiplication of the longitudinal and transverse radii by $\pi$. Results were expressed as mean $\pm$ S.E.M of lesion area (mm$^2$). A photographic register was also performed. The LS reduction percentage was calculated as follows: (LS0 $-$ LSx)/LS0 $\times$ 100, where LS0 = Initial lesion size and LSx = lesion size at each evaluation day.

### 2.6. Parasite Loads

DNA was extracted from parasites and tissue samples on days 0, 12, 25, and 60 p.t. using a DNeasy blood and tissue kit from Qiagen, following the manufacturer's instructions. A standard curve was performed by extracting DNA from *L. amazonensis* promastigote at a logarithmic phase and diluting it eight-fold, ranging from 1.0 to $1 \times 108$ parasites. The DNA purity index fell between 1.8 and 2.0. Three separate reactions of each concentration were conducted in triplicate. Each dilution point threshold cycle (Ct) was determined by calculating the Ct values' arithmetic mean and standard deviation. An average efficiency of 98.47% and a linearity coefficient r$^2$: 0.939 were obtained. The number of parasites per lesion was determined using quantitative PCR (qPCR) on a QuantStudio 5 Real-Time PCR System from Applied Biosystems (ABI). The primers used targeted a 170-pb region in the 18s rDNA gene (P1: 5′-CCAAAGTGTGGAGATCGAAG-3′/P2: 5′-GGCCGGTAAAGGCCGAATAG-3′). Each run included a standard curve and PCR reagents without DNA. Samples were evaluated in duplicate. To perform DNA amplification, 10 µL of PowerUp™ SYBR™ Green Master Mix from Thermo Fisher Scientific, Waltham, MA, USA, 9 µL of primer (0.5 µM), and 1 µL of DNA were used in a total volume of 20 µL. Thermal cycling conditions were 50 °C for 2 min and 95 °C for 2 min, 40 cycles at 95 °C for 15 s, and 60 °C for 1 min.

Ct values were obtained by measuring fluorescence emission. The mean Ct values were plotted against parasites' DNA serial dilutions with linear regression. The parasite loads of each amplicon were calculated using the ABI QuantStudio Design Analysis Software 1.4.3. The results were expressed as the mean $\pm$ S.E.M. of the number of parasites per lesion.

### 2.7. Histologic Analysis

Skin samples were collected from mice treated with 0.5% MTF hydrogel at different time points (0, 12, 25, and 60 days p.t.) with a sample size of 6 per time point. For the vehicle group, samples were collected from 4 mice on day zero and 2 mice each on days 12, 25, and 60. (Supplementary Table S2). Biopsies were fixed in 10% neutral formalin for 24 h, embedded in paraffin, and sectioned into 5 µm thick sections using a microtome. Dewaxed slices were stained with H & E and examined using microscopy. The epidermis was analysed for acanthosis, spongiosis, hypergranulosis, hyperkeratosis, parakeratosis, papillomatosis, and reactive keratinocytes. The dermis was examined for fibrosis, angiogenesis, inflammatory infiltrate, lymphocytes, plasma cells, polymorphonuclear cells, macrophages, eosinophils, giant cells, mast cells, granulomas, and amastigotes. The results were expressed in a categorical qualitative scale with crosses, as follows: (−) absence, (+) mild, (++) moderate, and (+++) severe [27].

### 2.8. Transmission Electron Microscopy (TEM)

Tissue samples were fixed with 3% glutaraldehyde in 0.1 M phosphate buffer pH 7.2 and post-fixed with 1% osmium tetroxide, washed in phosphate buffer, dehydrated in graded ethanol, infiltrated with 70 and 100% of resin and included in Epon (epoxy)-Araldite (Polysciences, Warrington, PA, USA). Ultrathin sections of 60 nm thickness were mounted over a grid, stained with uranyl acetate and lead citrate, and examined using a Zeiss EM 109 TEM. A description of the cellular structures was obtained after processing the samples at the Cellular Morphology Laboratory of the Colombian NIH.

### 2.9. Quantification of Proteins and Cytokines

Skin fragments of 5 mm were collected on days 0, 12, 25, and 60 p.t. and stored at −70 °C until protein and cytokine assays were performed. The skin was homogenised by using an Ultra-Turrax® (Ika, Campinas, Brazil) in 1.5 mL of lysis solution (PBS, pH 7.2, 0.2% Triton X-100, and protease inhibitors (SIGMAFAST™ Tablets, Sigma-Aldrich)), for four cycles of 15 s each. The homogenates were centrifuged at $10,000 \times g$ for 20 min at 4 °C, and supernatants were collected and stored at −70 °C. The protein concentration of the homogenates was determined using the Better BCA Protein Assay Kit, following the manufacturer's instructions (Bio Basic Inc., NY, USA). A standard curve was created by diluting bovine serum albumin and measuring protein concentration using sample O.D. values. The results were expressed as the mean $\pm$ S.E.M of protein concentration (mg/mL) in two independent experiments performed in triplicate. Simultaneous measurements of TNFα, IFNγ, vascular endothelial growth factor (VEGF), IL-4, and IL-10 in the homogenates were performed using Milliplex Map Mouse Cytokine/Chemokine Magnetic Bead Panel-Immunology Multiplex Assay; MCYTOMAG-60 K (Millipore Sigma, Burlington, MA, USA) according to the manufacturer's instructions. Cytokine concentrations were measured using a Luminex MAGPIX® instrument and xPONENT® software (Luminex, Austin, TX, USA). The results were expressed as the mean $\pm$ S.E.M of cytokine/mg of proteins in two independent experiments performed in triplicate. Skin homogenises from non-infected mice were used as a control. The Index of Cytokine Regulation infection (ICRi) was calculated by dividing the cytokines produced in CL lesions by those produced on healthy skin. The ICR treatment (ICRt) was calculated by dividing the cytokines produced in response to MTF treatment by those produced in response to vehicle treatment.

### 2.10. Statistical Analysis

One-way ANOVA and Tukey's multiple comparison tests were employed to analyse the progression in the size of the lesions within each group. Chi-square ($X^2$) analysis was used to determine associations between histopathological findings in subgroups treated with 0.5% MTF at different time points. Histopathological differences were not compared between the 0.5% MTF and vehicle groups due to the small sample size in the vehicle group (Supplementary Table S2, *n* = 2 on days 12, 25, and 60). The normality of the distribution of variables was verified using Kolmogorov–Smirnov tests. Given the non-parametric distribution of cytokine concentrations, Tukey's multiple comparisons and unpaired t-test were applied for the analysis. All statistical analyses were executed using GraphPad Prism® (Version 8.0; GraphPad Software, San Diego, CA, USA), with statistical significance acknowledged at *p* < 0.05.

## 3. Results and Discussion

### 3.1. CL Lesion Sizes and Parasite Loads before and after MTF Treatment

After 30 days, mice infected with *L. amazonensis* developed a nodular lesion that progressed to central ulceration after 60 days. In the group treated with MTF, the size of CL-lesions and parasite loads decreased. As the lesion healed, gradual closure of open ulcers and eventual scar formation or full re-epithelization occurred (Figure 1e). Lesion size (LS) measurements at 0, 12, 25, and 60 days p.t. were 34.22 ± 3.1 mm², 26.12 ± 2.8 mm², 4.49 ± 2.5 mm², and 2.47 ± 2.4 mm² (Figure 1a,c). The number of parasites per lesion was 304,862 ± 62,655, 140,005 ± 138,877, 3.52 ± 3.5, and 6976 ± 4687. Some parasites were detected 60 days p.t., but we did not test their viability. Percentages of lesion reductions were 23.67%, 86.88%, and 92.78%, while parasite reductions were 54.10%, 99.99%, and 97.70%. The group that received the vehicle showed an increase in LS at different evaluated times, which were 52.16 ± 3.7 mm², 61.78 ± 5.1 mm², 86.20 ± 8.2 mm², and 101.2 ± 18.12 mm², respectively, as shown in Figure 1b. There were no indications of weight loss, irritation, or pain symptoms during the treatment.

As with our findings, other authors reported decreased CL lesions and non-viable parasites after one month of follow-up while using 0.5% MTF in *L. amazonensis*-infected mice [28]. According to a study by Kavian et al. 2019 [29], applying liposomes containing 2% or 4% MTF topically twice daily for 4 weeks decreased lesion size and parasite loads in BALB/c mice infected with *L. major*. Viable parasites were scarce in the lesion and spleen after three months of follow-up [29]. Another study on BALB/c mice infected with *L. (L.) amazonensis* found that oral MTF doses above 10 mg/kg/day and 30 mg/kg/day (total MTF 6.3 and 19 mg) significantly reduced LS at the end of treatment, with no reactivation in LS or parasite numbers after eight months of follow-up [30]. However, a different study found that a higher MTF dose was ineffective and toxic, resulting in weight loss or death in *L. major*-infected BALB/c mice [31]. Our research shows that administering 5.3 mg MTF over 25 days of treatment significantly reduces lesion and parasite rates, with a 93% and 98% decrease, respectively. Notably, this final dose was lower than in previously reported experiments. We are concerned about the varying effectiveness of MTF in different experimental models. This inconsistency may be due to factors such as the involved parasite species, the host's genetic background, and the immune response elicited. Nonetheless, it is crucial to consider other aspects such as dosage, drug supplier, drug stability, and formulation type during clinical trials or use.

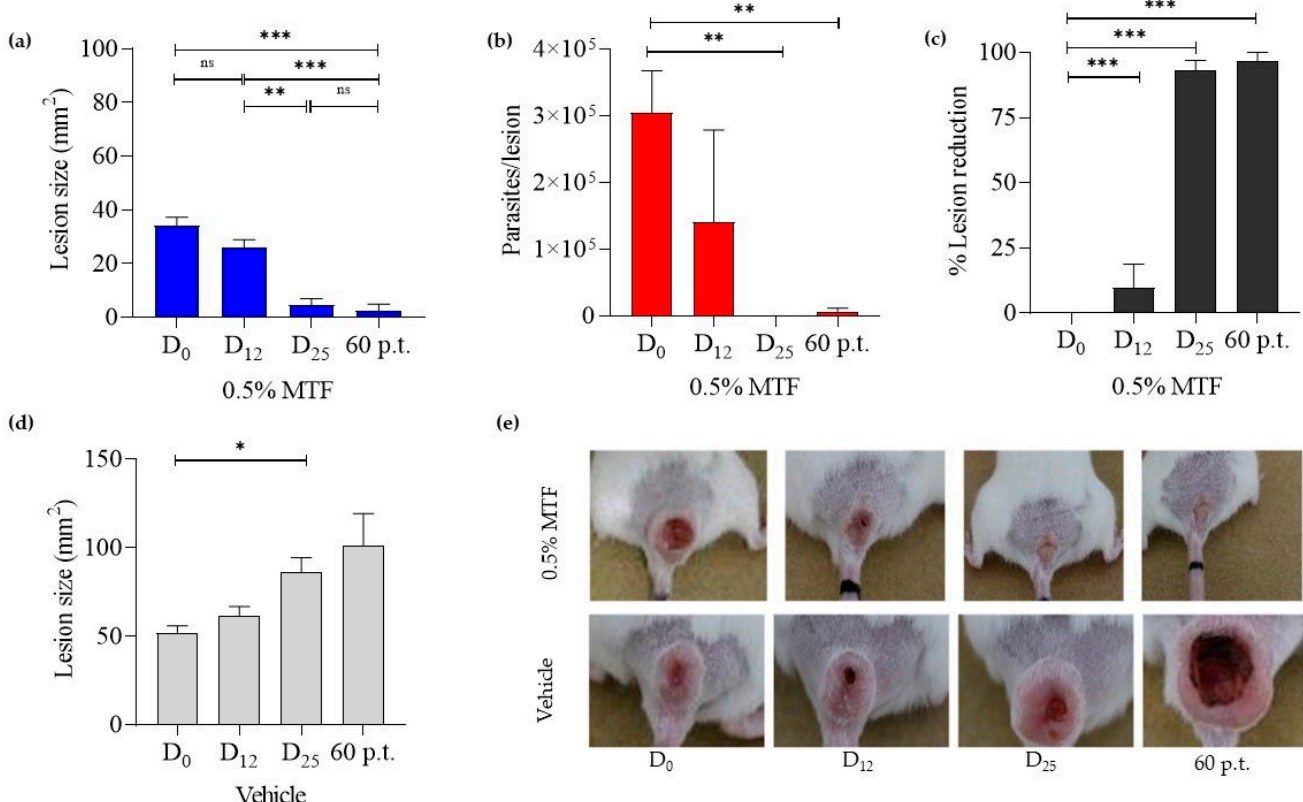

**Figure 1.** Monitoring lesion size and parasite loads in BALB/c mice infected with *L. amazonensis*, following topical MTF (0.5% MTF hydrogel) treatment at 0, 12, 25, and 60 days post treatment (p.t.). The figures show the drug effect on lesion size (**a**) and parasite loads (**b**), the % of lesion reduction induced by MTF (**c**), the effect of the vehicle treatment on lesion size (**d**), and the lesion images after MTF and vehicle treatment (**e**). The mean ± S.E.M. of lesion sizes after 0.5% MTF (*n* = 6) and vehicle (*n* = 4) and of the parasite load after 0.5% MTF (*n* = 3). Bars represent the standard error Turkey Multiple Comparison Test between different treatment days and p.t. * Significant differences $p < 0.05$, ** $p < 0.01$ ***, $p < 0.005$. D0: day 0; D12: day 12; D25: day 25; p.t.: 60 days post treatment.

### 3.2. Monitoring Histopathological Changes

On day 0, before treatments, the skin lesions displayed high intensity in nearly all the studied epidermal and dermal histopathological parameters (Figure 2, Supplementary Tables S1 and S2). Following the 0.5% MTF topical treatment, a decrease in the intensity of almost every evaluated one was observed (Figures 2 and 3), while no decrease was observed in the mice treated with the vehicle. Some exciting aspects induced by MTF treatment were the following: i. sustained increase in mast cells; ii. slight granuloma formation; iii. incomplete parasitological cure; and iv. transient papillomatosis (projection of dermal papillae above the skin surface). The role of mast cells in controlling or exacerbating leishmaniasis infections has been studied [32]. Their tissular localisation and potential abilities to release pre-existing cytoplasmic granules, including enzymes, histamine, NO, and TNFα, as well as Th1/Th2-type molecules after stimulation, indicate their pivotal role in both innate and adaptive antileishmanial immune responses [32]. The study showed that 0.5% MTF treatment led to an adequate response, resulting in increased mast cells and a small number of granulomas. Vehicle treatment did not increase mast cells or granuloma formation at 12, 25, and 60 days p.t. in the two mice analysed. In this specific analysis, a larger sample size of mice is needed to detect significant differences between vehicle subgroups or vehicle and MTF-treated groups. Mast cells have been implicated in early granuloma formation by releasing TNFα, which recruits and activates neutrophils to release chemokines, facilitating the recruitment of macrophages to the developing cutaneous granulomas [33]. In

BALB/c mice susceptible to *L. amazonensis* infection (see 0-day in Figure 3a), as well as in patients with anergic diffuse cutaneous leishmaniasis (DCL) caused by *L. amazonensis*, the infection results in nonhealing lesions characterized by diffuse inflammatory reaction and high parasite counts. In contrast, the presence of epithelioid cells, multinucleated giant cells, plasma cells, lymphocytes, mast cells or well-formed granulomas has been related to disease control [34]. A study showed that in over 50% of the researched cases of *L. tropica* and *L. major* infections (Old World CL), patients had fully formed granulomas and epithelioid histiocytes, leading to healing within 4.0 to 6.2 months [35]. Changes in histopathologic patterns (from susceptible to partially resistant) were also demonstrated after vaccination with solubilised *L. amazonensis* promastigote antigens. No vaccinated BALB/c mice showed extensive areas of vacuolated and parasitised macrophages (like fatty tissue); they showed coagulative but not fibrinoid necrosis. Nevertheless, in contrast, a focalised mixed-cell inflammatory reaction of small lymphocytes, macrophages, plasmocytes and eosinophils with diffuse fibrosis, giant cell formation, and scattered parasitised macrophages were observed on vaccinated mice. In addition, a granulomatous reaction with macrophages, epithelioid cells, multinucleated giant cells, focal fibrinoid necrosis disintegrating parasites, and a few cells within the necrotic areas was also observed [36]. The transient granuloma formation, especially in cases of *L. amazonensis*, could play an exciting role in MTF antileishmanial effectivity.

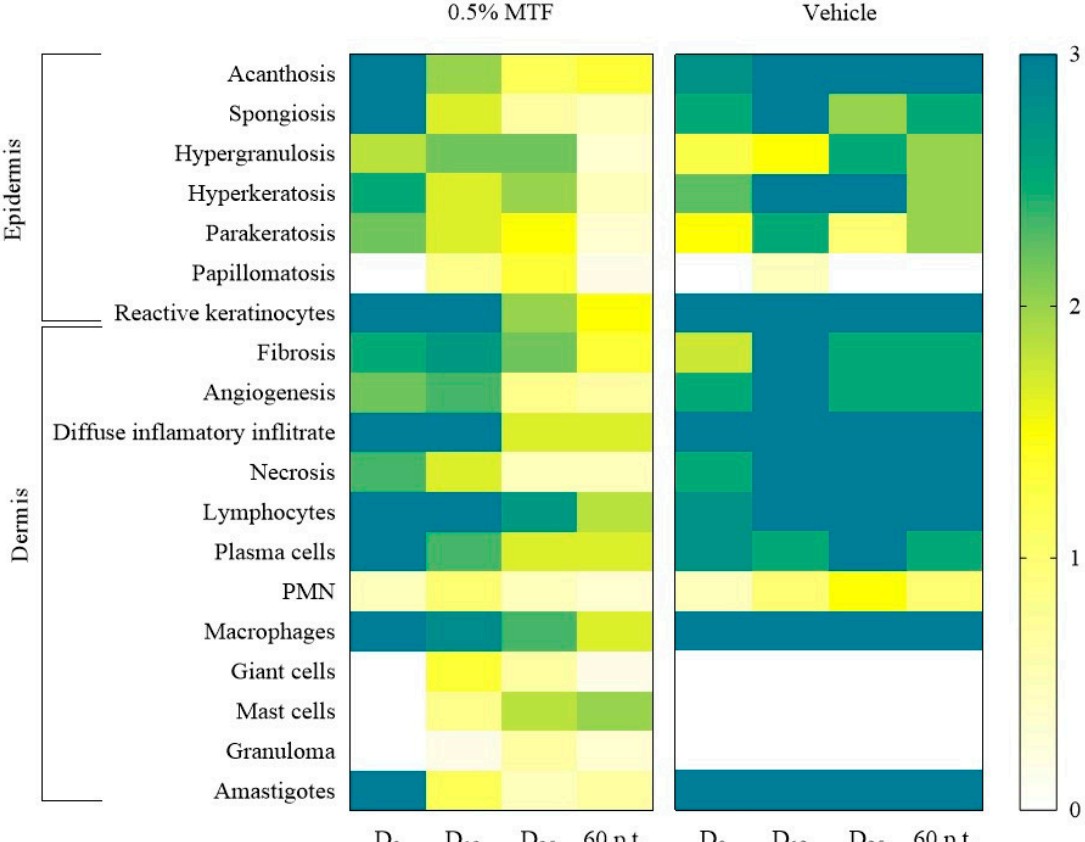

**Figure 2.** Monitoring histopathological parameters after 0.5% MTF and vehicle treatment in mice infected with *L. amazonensis*. A heat map was created using a gradient of colours based on the average intensity level (3+, 2+, 1+ and 0) of the histopathological features observed at day at 0, 12, 25, and 60 days post treatment. The *n* = 24 were treated with 0.5% MTF and *n* = 10 were treated with the vehicle. The changes were evaluated at day 0 ($D_0$), day 12 ($D_{12}$), day 25 ($D_{25}$), and 60 days p.t.

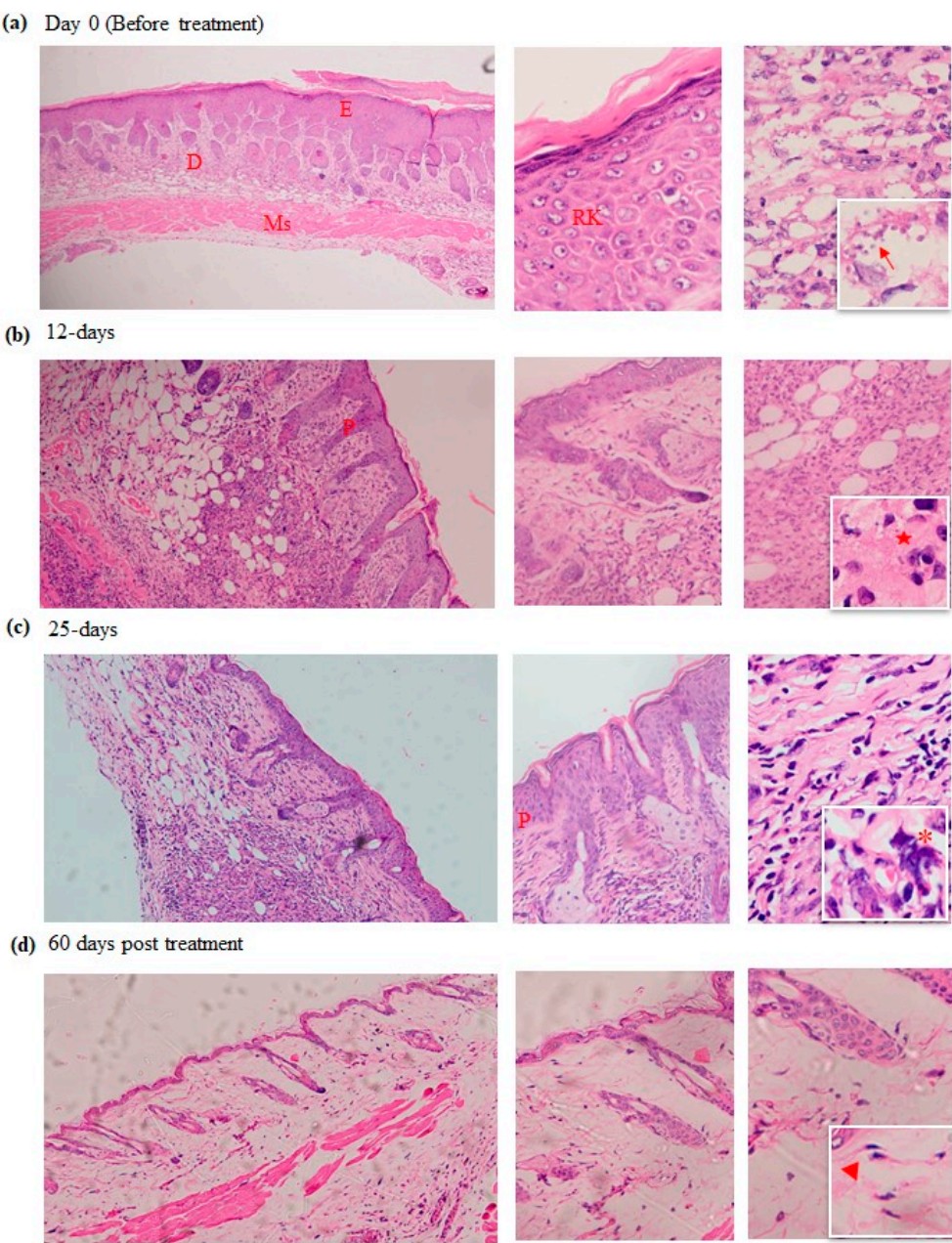

**Figure 3.** Histological study of cutaneous leishmaniasis skin sections taken before and at different time points during and after MTF treatment. The skin samples were stained with H&E, including samples taken at day 0 or before treatment (**a**), 12 days (**b**), 25 days (**c**), and 60 days after treatment with MTF hydrogel (**d**). Microphotographs were taken at different magnifications, including ×200, ×400, and ×1000. The third column highlights the presence of amastigotes (arrow), granuloma (star), mast cells (asterisk) and fibroblast cells (arrowhead). E, epidermis; D, dermis; Ms, muscle; RK, reactive keratinocytes; P, papillomatosis.

In our study, no complete parasite cure was observed. Our methods could not differentiate viable and non-viable parasites, potentially leading to underestimating treatment effectiveness. Usually, the therapeutic efficacy in CL is based on clinical parameters (complete lesion re-epithelialisation without reactivation after 90–360 days [37]. Our study observed a slight reactivation of parasite numbers at 60 p.t. Longer follow-up times are recommended, as complete parasite elimination in CL is rarely associated with esthetical cure, due to parasite persistence [38].

Finally, we observed a temporary papillomatosis, which may suggest some irritation during treatment. It is possible that MTF, which has surfactant properties, could affect skin integrity by causing protein denaturation, lipid depletion, and solubilisation in the stratum corneum [39]. However, in a previous study on healthy animals, a 0.5% *w/v* concentration of MTF did not show any signs of irritation or histological changes at the application site during a 14-day treatment period [27]. The slight irritation observed during treatment could be due to the drug properties, the 25-day treatment duration, or the use of DMSO at a concentration of 10% *v/v*. It has been shown in some models that even low concentrations of DMSO can irritate [40].

### 3.3. Ultrastructural Alterations in Infected Tissues during and after Treatment

On day 0, the macrophages showed different sizes of parasitophorous vacuoles containing numerous well-preserved amastigotes free or attached to the vacuole membrane. The amastigotes appeared with nuclear chromatin distributed normally, intact mitochondrial membranes, cell membranes, and kinetoplasts (Figure 4a,b). No parasites were observed outside of the vacuole. After 12 days of 0.5% MTF-hydrogel, conserved macrophages exhibited multiple cytoplasmatic vacuoles without amastigotes in the fields examined. Most of these vacuoles contained electron-dense lipid-like material, while others held residual parasite structures. In addition, fibroblasts, collagen fibres, and eosinophils were observed (Figure 4c,d). After completing 25 doses of the treatment, the cells displayed numerous vacuoles in their cytoplasm. Some of these vacuoles contained highly particulate electron-dense material, while others held a smaller amount of homogeneous electron-dense material. No amastigotes were detected in the sample, as seen in Figure 4e,f.

As with our findings, tissular vacuolated macrophages without parasites were observed after topical application of AmB-MTF liposomes twice daily for four weeks on CL lesions from BALB/c mice infected with *Leishmania (Leishmania) mexicana* [41]. Ultrastructural changes in cutaneous lesions after oral MTF treatment (20 mg/kg per day for 20 days) were studied in BALB/c mice infected with *L. major* [42]. On day 0, conserved parasites were found outside and inside the vacuoles. At day 20 (end of treatment), the amastigotes showed signs of plasma membrane disintegration, partial disappearance of subpellicular microtubules, and the absence of a nuclear envelope with abnormal heterochromatin distribution. Additionally, non-membrane-bound cavities were observed, while the parasite mitochondria remained preserved. Condensation of chromatin and the nucleus (pyknosis), which could be related to both apoptotic and necrotic cell death, was observed in some macrophages [42]. Studies in vitro on *L. amazonensis*-infected macrophages treated with edelfosine, an alkyl-lysophospholipid, revealed several changes in parasites, such as cytoplasmic vesicles, swelling of the mitochondria with a disruption in the inner-membrane organisation, multinucleated parasites, parasitophorous vacuoles with intense membrane vesiculation, and parasite debris [43]. Furthermore, MTF had a direct impact on *Angomonas deanei* (a trypanosomatid protozoan) ultrastructure, including the development of blisters and detachment of the plasma membrane, as well as swelling of the mitochondria with enlarged ridges and significant cell vacuolisation [44]. The ultrastructural analysis of this work using monotherapy with MTF showed the damage or fragmentation of the cell membranes of the parasites, possibly induced by better action in the oxidative response of the macrophage during treatment, and which, in addition to cell apoptosis, indicates that the growth and parasitic development observed in the decrease in the parasite load is inhibited.

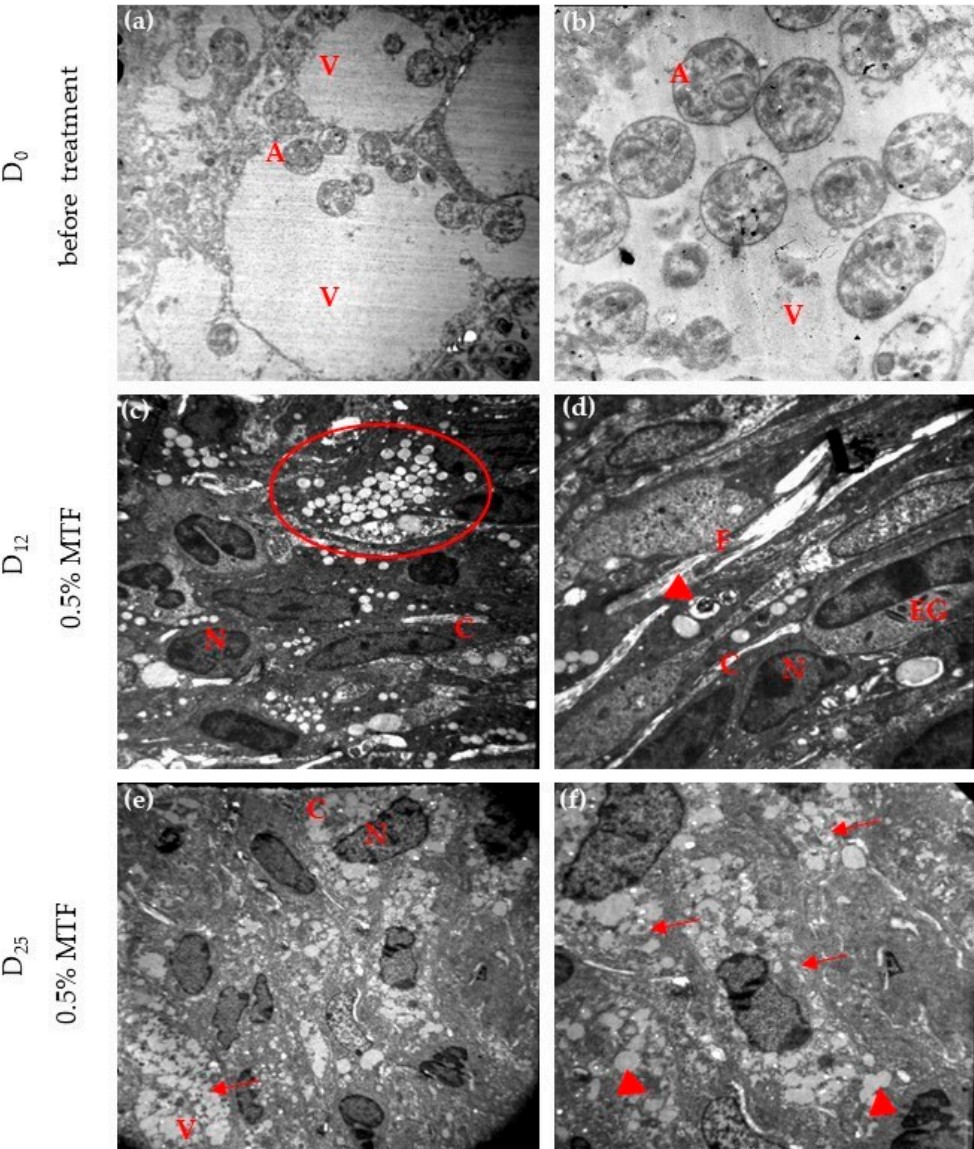

**Figure 4.** Monitoring ultrastructural alterations during 0.5% MTF treatment. Ultra-thin skin sections at different magnifications in a TEM were observed before (day $D_0$) and 12 and 25 days of MTF treatment. The photographs showed free amastigotes or amastigotes attached to the membrane macrophage at $3000\times$ (**a**) and vacuoles and intact amastigotes inside one vacuole at $4400\times$ (**b**). At day $D_{12}$, multiple intracellular vacuoles with lipid-like material (red circle) at $3000\times$ (**c**) and vacuoles with parasitic residues (arrowhead) at $7000\times$ (**d**). At day $D_{25}$, end of treatment, macrophages with many vacuoles in their cytoplasm, several with a particulate electron-dense material (arrow) $3000\times$ (**e**) and vacuoles with homogeneous material (arrowhead) and particulate electron-dense material (arrow). No amastigotes were identified at $7000\times$ (**f**). A: amastigote, V: vacuole, N: nucleus, C: cytoplasm, F: collagen filaments, EG: eosinophilic granules.

### *3.4. In Situ Cytokine Regulation*

3.4.1. Cytokine Regulation by Infection with *L. amazonensis*

After 4 to 6 weeks of infection, areas of CL lesions displayed an average size of 40 mm$^2$. Lesion lysates exhibited levels of IL-4, IFN$\gamma$, IL-10, TNF$\alpha$, and VEGF of $38.71 \pm 6.2$, $12.55 \pm 3.1$, $11.52 \pm 2.3$, $4.35 \pm 0.5$, and $9.28 \pm 2.3$ pg/mg protein, respectively. The levels of IL-4, IFN$\gamma$, IL-10, TNF$\alpha$, and VEGF of non-infected skin (healthy skin) showed levels of $0.63 \pm 0.3$, $5.93 \pm 1.1$, $8.38 \pm 1.6$, $0.46 \pm 0.26$, and $20.56 \pm 1.9$ pg/mg protein, respectively (Figure 5a). The IL-4 and TNF$\alpha$ levels were significantly higher in CL lesions than in

non-infected skin ($p < 0.001$), with ICR values of 61.4 and 9.5, respectively. IFNγ levels were slightly higher (ICR 2.1); IL-10 showed a similar level, with an ICR value of 1.4, while VEGF was found to be downregulated in cutaneous lesions, with an ICR value of 0.45 (Figure 5b). In similar experimental models, i.e., for L. major infection, BALB/c mice are susceptible and produce high levels of IL-4 and IL-10 (Th2 response), while C57BL/6 mice are resistant, and produce high levels of IFNγ but no IL-4 and IL-10 (Th2 response) [6]. For *L. amazonensis* infection, as with other NW-*Leishmania* species, both strains of mice are susceptible, and the Th1/Th2 dichotomy is not always observed. In a study on C57BL/6 mice, it was found that chronic ulcerative lesions were linked to high levels of proinflammatory cytokines mRNA, such as IL-12, TNFα, and IFNγ, as well as iNOS during the late phase of infection [45]. In our research, we observed an increase in TNFα levels (ICR 9.5), increased levels of IFNγ (ICR 2.1), no change in IL-10 levels, and, like *L. major* infection in susceptible BALB/c, a rise in IL-4 (ICR 62.4).

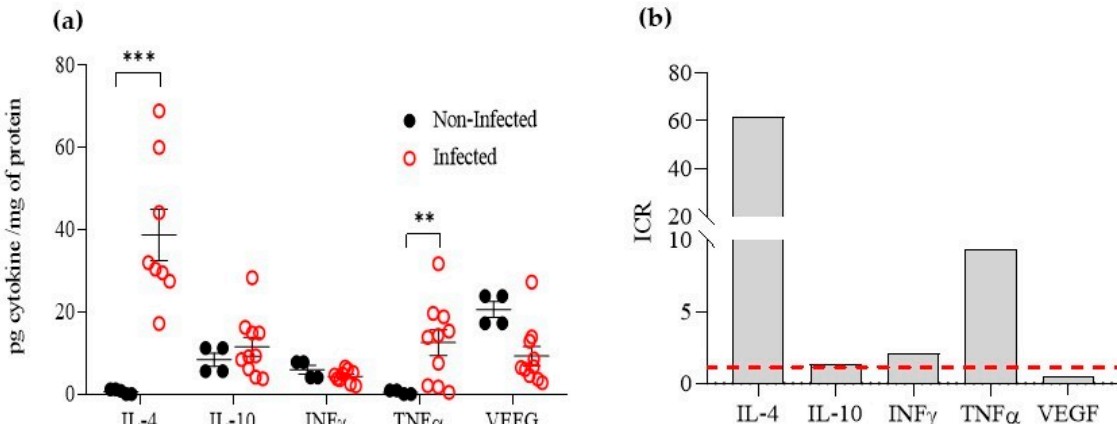

**Figure 5.** Cytokine regulation in cutaneous leishmaniasis (CL) lesions caused by *L. amazonensis* in BALB/c mice. Cytokine concentration on non-infected and infected skin (**a**). Indexes of cytokine regulation (ICRs). The red dashed line indicates ICR = 1 i.e., when infected and normal skin cytokines levels were equal. (**b**). The ICRs were calculated by dividing the cytokine levels in the CL lesion by those in non-infected skin. The data were analysed using the unpaired *t*-test to determine significant differences. ** $p < 0.01$, *** $p < 0.001$.

### 3.4.2. IL-4, TNFα, and VEGF Levels Decreased after MTF Treatment

Before treatment, on day 0, the cytokine levels in the tissues of the 0.5% MTF hydrogel or vehicle experimental groups were similar, with ICRs of around 0.9 and 1.1. At 12, 25 and 60 days p.t., the ICRs (0.5% MTF-hydrogel group/vehicle group cytokines) for IFNγ and IL-10 increased and decreased slightly, displaying ICR values of 0.48, 1.12 and 0.70 for IFNγ and ICRs of 0.93, 0.79 and 1.48 for IL-10, at 12, 25 and 60 days pt. There were no statistically significant differences in IFNγ and IL-10 levels between the MTF and vehicle treated groups (Figure 6f,g). In the MTF group, the levels of IFNγ were 8.94, 11.87, and 14.02 pg/mg protein, while the levels of IL-10 were 10.23, 9.94, and 12.29 pg/mg protein, respectively, at the different evaluated times. In the vehicle group, the levels of IFNα were 18.63, 10.61, and 20.02 pg/mg protein, while the levels of IL-10 were 10.97, 12.57, and 8.32 pg/mg protein (Figure 6b,c).

The picture regarding the tissular regulation of IL-4, TNFα, and VEGF through MTF treatment was different. The ICRs decreased. Values of ICRs were 0.41, 0.07, and 0.06 for IL-4; ICRs were 0.75, 0.24, and 0.13 for TNFα; and ICRs were 0.42, 0.42, and 0.21 for VEGF, at 12, 25, and 60 p.t., respectively (Figure 6f,g). Statistical differences were demonstrated for IL-4 and TNFα (at 25 and 60 days p.t.) and VEGF (at day 60 p.t.). The levels of cytokines for MTF versus the vehicle group were 15.95, 3.25, and 3.38 pg/mg protein versus 48.69, 44.21, and 53.64 pg/mg protein for IL-4 (Figure 6a); 2.84, 0.99, and 0.86 versus 3.78, 4.10, and 7.20 pg/mg protein for TNFα (Figure 6d); and for VEGF, 2.96, 5.78, and 4.37 versus

7.08, 13.68, and 20.94 pg/mg protein (Figure 6e). The results showed that IL-4, TNFα, and VEGF levels decreased after MTF treatment.

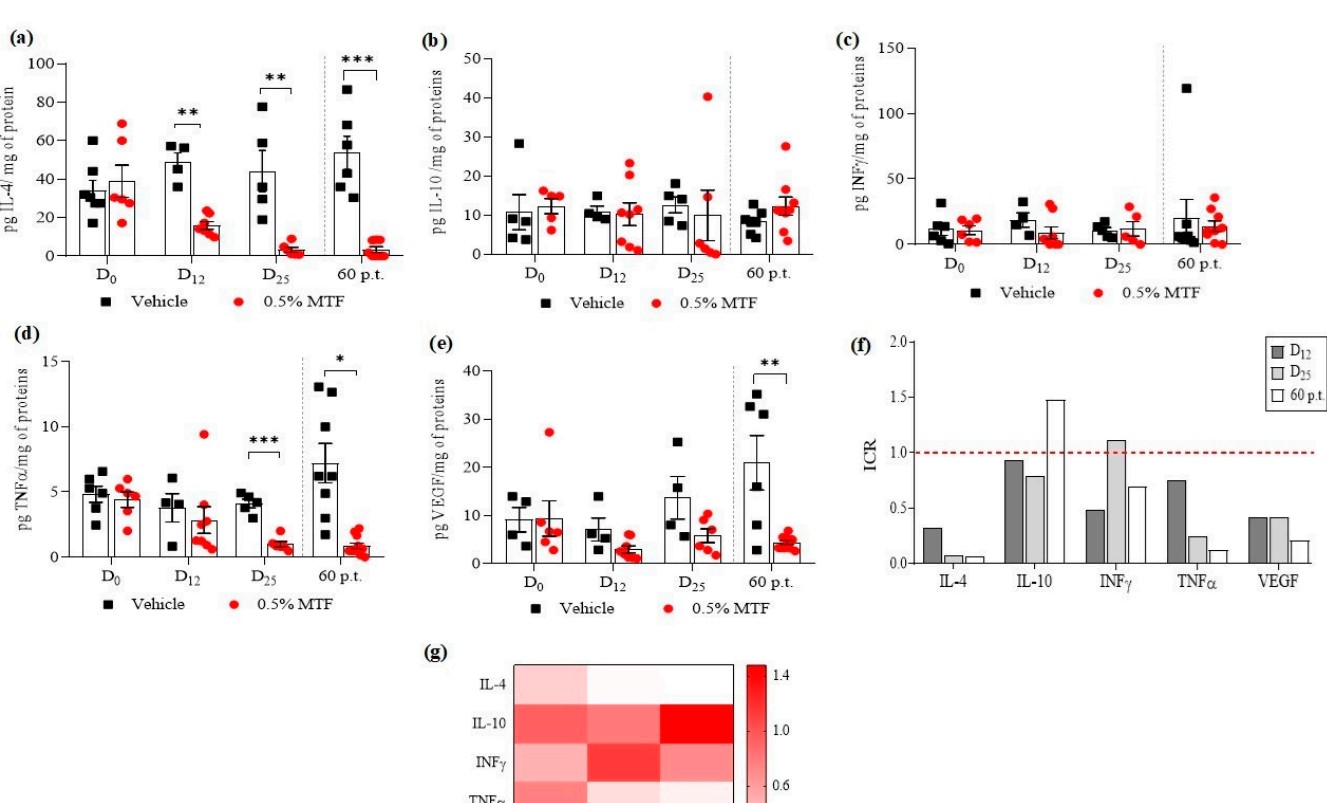

**Figure 6.** Monitoring cytokines released after 0.5% MTF and vehicle topical treatment at different intervals (Day $D_0$; $D_{12}$, $D_{25}$, and 60 days p.t.). The figures show the release of IL-4 (**a**), IL-10 (**b**), INFγ (**c**), TNFα (**d**) and VEGF (**e**) Treatment days were separated from post-treatment days by the grey dashed line. The mean ± S.E.M. of cytokine concentration produced by mice treated with 0.5% MTF (*n* = 6) and vehicle (*n* = 4). Differences between groups were determined using Tukey's Multiple Comparison Test. Significant differences: * $p < 0.05$, ** $p < 0.01$, *** $p < 0.001$. The indexes of cytokine regulation (ICRs) induced by MTF compared to the vehicle. The red dashed line indicates ICR = 1 i.e., when cytokine levels after MTF and vehicle treatments were equivalent; (**f**) A heat map based on the ICRs (0 = white, maximal decrease: 1–1.5 = red, similar o minimal difference compared with the vehicle), at 12, 25, and 60 days p.t. (**g**).

The cytokine response to MTF therapy has been demonstrated in various in vitro, in vivo, and clinical trials, especially in leishmaniasis, due to *Leishmania donovani* [18]. In a patient with PKDL, transcripts of IFNγ, IL-10, and TNFα genes were present in the lesion tissue before treatment. However, after MTF treatment, a significant increase in IFNγ transcripts and a decrease in TNFα and IL-10 transcripts were observed [46]. In another study, after four months of MTF treatment at a dose of 100 mg/day, PKDL patients (*n* = 12) showed increased TNFα and decreased IL-10 levels, with levels of IL-4 remaining roughly the same. Additionally, stimulated blood mononuclear cells (PBMC) from patients treated with MTF displayed higher TNFα levels and lower IL-10 levels than untreated patients [47]. Another study showed that levels of IL-10 decreased in the treated group, and it was correlated with the disappearance of dermal lesions and the absence of parasite load [48]. In an in vitro study using *L. donovani*-infected THP-1 cells or PBMC from VL patients at different time points of MTF treatment, treatment with MTF or stimulation with leishmanial antigen displayed an increase in proinflammatory cytokines (IL-12, TNFα

and IFNγ), NO and a decrease in IL-10 and TGFβ [49]. Our work presents a different view compared to previous studies. We found that levels of IFNγ and IL-10 were similar, while levels of TNFα were decreased in CL lesions after MTF treatment. These differences may be attributed to several factors, including the specific clinical manifestations under investigation (VL, PDKL vs. CL), the types of samples utilized (plasma, RNA from tissues, and cell culture supernatants vs. mouse skin tissue lysate, as used in our study), and the methods employed for quantifying cytokine levels. There are limited studies regarding the impact of MTF on immunomodulation in NW-CL in humans. In a clinical trial involving patients with *L. (V.) braziliensis* infection, PBMCs were collected before and during treatment with oral MTF. No significant differences were observed in the IFNγ, TNFα, and IL-10 levels before or during therapy [21]. In a study using *L. infantum*-infected hamsters, splenocytes stimulated by antigens and obtained 15 days after a 28-day oral treatment with MTF resulted in an increase in the total number of T-CD4 splenocytes that produced IFNγ and TNFα. However, there was a decrease in the total number of splenocytes that produced IL-10 [50]. As for the downregulation of VEGF, the vascular remodelling seen in lesions could explain associations between VEGF and CL. This remodelling generally occurs within a hypoxic and inflammatory microenvironment. Certain *Leishmania* species, including *L. major*, *L. amazonensis*, and *L. donovani*, have been linked to the direct activation of the hypoxia-inducible transcription factor 1α (HIF-1α) within the skin. This activation induced cytokines and the release of growth factors like VEGF by macrophages [51]. In our study, treatment with MTF resulted in a decrease in both VEGF and inflammatory infiltrate in the tissue. The decline in VEGF levels observed after MTF treatment contrasts with the continuous rise in VEGF levels observed after vehicle treatment.

## 4. Conclusions

The topical application of 0.5% MTF-hydrogel on *L. amazonensis*-infected mice resulted in positive in situ results from the early stages of usage. The MTF can directly target parasites and indirectly downregulate the release of IL-4, TNFα, and VEGF. It can also increase the number of mast cells and promote a mild granuloma reaction during different evaluation periods. Before treatment, BALB/c mice exhibited a mixed Th1/Th2 cytokine response in the CL lesion in response to *L. amazonensis*. However, during and after treatment and in the healing process, the immune response after MTF treatment was characterized by decreased IL-4, TNFα, and VEGF production. After a 60-day follow-up period, a few parasites persisted on the lesions. Additional trials should evaluate higher MTF doses or combined therapies.

**Supplementary Materials:** The following supporting information can be downloaded at: https://www.mdpi.com/article/10.3390/scipharm91040054/s1, Figure S1: Identification of *Leishmania* species in this study using (a.) HSP70-PCR, (b.) ITS1-PCR.; Table S1: Characteristics and intensity of the histopathological variables studied in lesions from mice with CL treated topically with 0.5% MTF-hydrogel. Table S2: Characteristics and intensity of the histopathological variables studied in lesions of mice with CL treated topically with the vehicle without MTF.

**Author Contributions:** Conceptualization, L.F.N., J.C.M. and P.E.; methodology, L.F.N., J.C.M. and P.E.; formal analysis, L.F.N. and P.E.; investigation, L.F.N., J.C.M. and P.E.; resources, L.F.N. and P.E.; writing—original draft preparation, L.F.N. and P.E.; writing—review and editing, L.F.N., J.C.M. and P.E.; project administration, L.F.N. and P.E.; funding acquisition, L.F.N. and P.E. All authors have read and agreed to the published version of the manuscript.

**Funding:** This research was funded by the Colombian Ministry of Science, Technology, and Innovation, MINCIENCIAS-Colfuturo grant 757–2016 for national PhD students, MINCIENCIAS grant 2022–0645, and the Industrial University of Santander.

**Institutional Review Board Statement:** The animal study protocol was approved by the Ethics Committee (CIENCI) of the Industrial University of Santander, Bucaramanga, Colombia (Acta 23 date of approval: December 2019).

**Informed Consent Statement:** Not applicable.

**Data Availability Statement:** Data are contained within the article.

**Acknowledgments:** We would like to thank Jorge Osorio and Angelica M. Vera for their support in some experiments.

**Conflicts of Interest:** The authors declare no conflict of interest.

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
