# Peer review of "Monitoring Cutaneous Leishmaniasis Lesions in Mice Undergoing Topical Miltefosine Treatment"

_scipharm, doi:10.3390/scipharm91040054_

Round 1

Reviewer 1 Report

Comments and Suggestions for Authors

This article presents interesting research on the topical application of miltefosine in cutaneous leishmaniasis in an experimental mouse model. Topical miltefosine has been used previously in in-vitro and in-vivo research, but novel data from this article include interleukin tissue measurements and measurements of ultrastructural changes in the inflammatory cells and infected tissue.

Materials and methods require some major corrections: 40 mice were infected with L. amazonensis and, after 60 days of well-developed skin lesions, "randomly divided into two experimental groups, one treated with 0.5% MTF hydrogel (n = 24) and the other with vehicle (n = 16). Mice were sacrificed at four time points: Day 0 (before treatment), Day 12 and Day 25 (during treatment), and Day 60 p.t." (lines 108-110).

A look at the supplemental materials (Tables S1 and S2) shows that in S1 the results of 24 mice treated with MTF are presented, but in S2 only 10 mice are presented. The authors should explain what happened to the remaining 6 mice treated with vehicle This should be explained in the Materials and Methods section. In addition, it is not entirely clear to the reader that for each study time point (D0;D12;D25;D60 namely), only "one quarter" of the respective group of mice was investigated. Therefore, biopsies were not taken from the entire group at each time point. Statistics were therefore obtained from 6 mice at each time point in the MTF group and from an even smaller number of mice at each time point in the vehicle group (4 mice at day zero; 2 mice at day 12; 2 mice at day 25; and 2 mice at day 60; Table S2). This is a very small number of mice tested at each time point, and it is suggested that the authors explain how this was resolved with the statistics An explanation is therefore suggested in Materials and Methods under Statistics (lines 184-192).

Line 306: Typo: The word "exanimated" should be replaced by "examined"

Author Response

We appreciate the time and effort you took to review our manuscript. Your valuable comments will undoubtedly enhance the quality of our work. Thanks

Response:  Regarding your answer, yes, on days 12, 25, and 60, only two mice were tested. Statistical analysis was performed only within subgroups treated with 0.5% MTF

We added:

2.4.         Infection and Treatment

The mice (n = 40) were infected by subcutaneous injection of 5 x 105 stationary phase L. amazonensis promastigotes suspended in 100 µL of PBS, pH 7.2, in the shaven rump above the tail. When the lesions were ˃ 20 mm2, they were randomly distributed into two experimental groups, one treated topically with 0.5% MTF hydrogel and the other with a vehicle directly over the CL lesion. The group treated with MTF had a sample size of n = 24 mice, distributed in 4 subgroups of n = 6 mice each. On the other hand, the vehicle group had a sample size of n = 16 mice, distributed in 4 subgroups of n = 4.

2.7. Histologic Analysis

Skin samples were collected from mice treated with 0.5% MTF hydrogel at different time points (0-, 12-, 25-, and 60-day p.t.) with a sample size of 6 per time point. For the vehicle group, samples were collected from 4 mice on day zero and 2 mice each on days 12, 25, and 60 (Table S2).

2.10 Statistical Analysis

Chi-square (X2) analysis was used to determine associations between histopathological findings in subgroups treated with 0.5% MTF at different time points. Histopathological differences were not compared between the 0.5% MTF and vehicle groups due to the small sample size in the vehicle group (Table S2, n = 2 on days 12, 25, and 60).

Figure 2.

A heat map was created using a gradient of colours based on the average intensity level (3+, 2+, 1+ and 0) of the histopathological features observed at day at 0-, 12-, 25-, and 60 days post-treatment (p.t.). The n = 24 mice were treated with 0.5% MTF and n = 10 were treated with vehicle.

3.2. Monitoring Histopathological Changes

We modified the sentence written in the result and discussion,

Compared with the vehicle, an effective MTF-induced response comprised increased mast cells and small and few granulomas.

Was modified as:

The study showed that 0.5% MTF treatment led to an adequate response, resulting in increased mast cells and a small number of granulomas. Vehicle treatment did not increase mast cells or granuloma formation at 12, 25, and 60 days p.t. in the two mice analysed. In this specific analysis, a larger sample size of mice is needed to detect significant differences between vehicle subgroups or vehicle and MTF-treated groups.

Justification: Unfortunately, we don't have any further histopathological data for vehicle treatment on days 12 and 25. Histopathological outcomes by topical vehicle treatment (Carbopol 1%, DMSO 10%) after a 60-day follow-up have been described in our lab in other studies, and similar results have been displayed. Additionally, vehicles treated mice behaviour as untreated mice, especially for dermal studied parameters. However, this data was not included.

We used the six missing samples mainly for measuring parasite loads and cytokine levels, so we opted not to preserve them in formalin for use in histopathology tests. Unlike the histologic categorical variables, the quantitative variables (parasite loads and cytokine levels) were standardized for the first time in our laboratory.

Comment 2: Line 306: Typo: The word "exanimated" should be replaced by "examined"

Response: OK, was replaced, thanks

Reviewer 2 Report

Comments and Suggestions for Authors

This study describes de effect of a hydrogel for the treatment of cutaneous lesions caused by Leishmania amazonensis. A comprehensive evaluation of those lesions was carried out until day 85. Miltefosine is an important therapeutic agent for visceral leishmaniasis, and its use for cutaneous leishmaniasis is also worthy of investigation.

Under this reviewer’s criteria, the manuscript deserves being considered for publication pending several adaptations in order to increase its value for the readers.

Abstract – please provide explanations for the meaning of MTF and TEM – the same for the cytokines

Keywords – display alphabetically

Line 37 – write Leishmania spp.

Explain the meaning of all cytokines at their first use

Line 51 – post-kala-azar dermal leishmaniasis occurs after Old World visceral leishmaniasis – not CL

Line 71 – replace effectiveness with efficacy

Write out Leishmania when presenting a new species

Line 89 – not clear what (L.) and (V.) stand for

Line 106 – which criteria have determined a sample sizer of 40 mice?

If Results and Discussion are presented together, perhaps this manuscript should be revised as a short communication

Line 195 – use L. amazonensis – instead of L. (L.) amazonensis

Line 408 – PKDL should be abbreviated at its first use, i.e. line 51

Comments on the Quality of English Language

Minor editing of English language required.

Author Response

We appreciate your time and effort in reviewing our manuscript. Your valuable feedback will undoubtedly enhance the quality of our work.

Reviewer 2: Thank you for noticing the error. We have made a correction and changed "exanimated" to "examined." We appreciate your attention to detail.

Comment 1: Abstract – please provide explanations for the meaning of MTF and TEM – the same for the cytokines.

Response: OK, thanks

Comments 2: Keywords – display alphabetically.

Response: OK, they have been rearranged alphabetically.

Comments 3: Line 37 – write Leishmania spp.

Response: This has been corrected to "Leishmania spp."

Comments 4:  Explain the meaning of all cytokines at their first use.

Response: OK, we did. Thanks

Comments 5: Line 51 – post-kala-azar dermal leishmaniasis occurs after Old World visceral leishmaniasis – not CL

Response: Sorry, it was a mistake. It has been corrected by Old World visceral leishmaniasis.

Comments 6: Line 71 – replace effectiveness with efficacy.

Response: We did, thanks

Comment 7: Write out Leishmania when presenting a new species.

Response: Each new species is now introduced with the full "Leishmania" term for clarity.

Comments 8: Line 89 – not clear what (L.) and (V.) stand for to avoid confusion.

Response: We added (L.) and (V.) to name Leishmania species, especially for New World cutaneous leishmaniasis, where both subgenera are prevalent. They correspond to the subgenus Leishmania and Viannia, respectively, as mentioned in line 31. We left (L.) and (V.) only the first time they were introduced.

Comments 9: Line 106 – which criteria have determined a sample sizer of 40 mice?

Response: The group treated with MTF had a sample size of n = 24 mice, distributed in 4 subgroups of n = 6 mice each. On the other hand, the vehicle group had a sample size of n = 16 mice, distributed in 4 subgroups of n = 4. Each subgroup represents the mice sacrificed at the different evaluated times The appropriate sample size (n) for our study was determined using G Power software (Statistical Power Analyses 3.1.9.4 for Windows), based on previously obtained values from our lab for standardizing tissue parasite loads and cytokine, with a power of 80% and alpha of 0.05.

Comments 10: If Results and Discussion are presented together, perhaps this manuscript should be revised as a short communication.

Response: We appreciate the suggestion regarding our manuscript. We believe our manuscript's structure sufficiently covers the required elements and provides a thorough analysis, according to Scientia Pharmaceutica instructions. The original article allowed Discussion combined with Results with a minimal word count of 4000. We have exceeded the maximum word count of 2000 for the "Communication" document.

Comments 11: Line 195 – use L. amazonensis – instead of L. (L.) amazonensis

Response: OK, it was corrected.

Comments 12: Line 408 – PKDL should be abbreviated at its first use, i.e. line 51

Response: PKDL is now abbreviated at its first use on line 53.

Comments on the Quality of English Language: Minor editing of English language required.

Response: We improved it in some parts and hope it is now better.